# Hidden order in dielectrics: string condensation, solitons, and the charge-vortex duality

Sergei Khlebnikov

*Department of Physics and Astronomy, Purdue University, West Lafayette, IN 47907, USA*

## Abstract

Description of electrons in a dielectric as solitons of the polarization field holds promise as a method for computer simulations of the dynamics of excited states. For that description to be realistic, the interaction between the solitons (prior to their coupling to electromagnetism) must be short-ranged. We present an analytical study of the mechanism by which this is achieved. It is unusual in that it enables screening of electrically neutral soliton cores by polarization charges. We also argue that the structure of the solitons allows them to be quantized as either fermions or bosons. At the quantum level, the theory has, in addition to the solitonic electric, elementary magnetic excitations, which give rise to a topological contribution to the magnetic susceptibility. We suggest that this effect may be enhanced in radiatively excited states of small crystals.

# 1   Introduction

Recent years have brought attention to ordered phases of matter in which the order cannot be described within the familiar paradigm of symmetry breaking and which therefore require complementary concepts. Perhaps the most familiar example of such an order is provided by a type-II superconductor in three dimensions: this system does not have a local order parameter, yet that does not preclude the existence of a continuous phase transition [1]. By itself, then, the view that a hidden order, not associated with condensation of any local field, can exists in a condensed-matter system is hardly in doubt; what is less well established is how common this phenomenon is.

In the example above, an important role is played by the topological defects—the vortex lines—and the fact that in the superconducting phase, where these lines can be described semiclassically, the interaction between them is short ranged. This is in contrast to long-range interactions mediated by the Nambu-Goldstone bosons in theories with spontaneously broken continuous global symmetries, as for example in the Skyrme model [2], where the interaction energy of two solitons decays as $1/r^3$ at large distances [3].

One may wonder if the property just discussed for the special case of a superconductor, namely, the existence of topological defects (solitons) with exponentially decaying interactions is in fact a general characteristic of a whole class of hidden orders. (We will continue to refer to these solitons as "defects," despite the absence of a conventional long-range order, because they are essentially screened versions of more conventional defects: monopoles or vortices.)

In the present paper, we consider from this perspective the description of electrons and holes in a dielectric as solitons of the polarization field. We proposed this description in [4] as a possible means to enable computer simulations of excited states containing multiple electrons and holes, in the regime where the available phase space is large enough for the solitons representing these particles to be treated classically. The existence of unconfined solitons in this model relies on the invariance of the static energy functional with respect to adding closed strings of quantized electric flux. This makes it similar in spirit to the models of string-net condensation, proposed as a description of a non-symmetry-breaking order in Refs. [5, 6]. Here, our focus is on developing an analytical argument that would allow us to establish the form of the soliton interactions.

|  | Magnetic case (superconductor) | Electric case (dielectric) |
|---|---|---|
| Angular variable | phase $\theta$ | emergent gauge field $\boldsymbol{\psi}$ |
| Screening field | vector $\mathbf{A}$ | scalar $\chi$ |
| Derivative object | covariant derivative $\boldsymbol{\pi} = \nabla\theta - \mathbf{A}$ | polarization vector $\boldsymbol{p} = \nabla \times \boldsymbol{\psi} + \nabla\chi$ |
| Quantized charge | magnetic flux $= \int \nabla \times \mathbf{A}$ | electric charge $= -\int \nabla^2 \chi$ |

Table 1: Comparison of magnetic screening in a phase-only model of superconductor and electric screening in the solitonic theory of dielectrics in three dimensions. In either case, we focus on the screening of a topological defect in the corresponding angular variable due to the presence of another ("screening") field.

We find that the interactions decay exponentially (prior to coupling of the solitons to electromagnetism), as a consequence of a screening mechanism that allows an uncharged soliton core to source a polarization cloud with the total charge of unity. The mechanism is distinct from (in a sense, even opposite to) the Debye screening in plasma, wherein a nonzero core charge produces a configuration of the overall charge zero. Rather, it is similar to the screening of the interaction between vortices in a superconductor by the vector field, where a defect in the order parameter sources a quantized magnetic flux. Indeed, we will see that in two spatial dimensions, at the level of static configurations, the correspondence between the two mechanisms becomes precise (a change of variables). In three dimensions, that is no longer so; indeed, the defects in a superconductor and in an insulator even have different dimensionalities. Nevertheless, some parallels remain. A comparison of the two screening mechanisms in three dimensions is presented in Table 1.

The similarity with the magnetic screening in superconductors becomes especially close in the limit when the size $\mu^{-1}$ of the polarization cloud of the soliton (the analog of the London depth in a superconductor) is large in lattice units. We refer to this limit as an easily polarizable medium. In this case, the soliton core, defined as the region at distances $r \ll \mu^{-1}$ from the soliton center, is practically uncharged. It is described [4] by a gauge theory—the lattice electrodynamics [7] of an emergent gauge field. At these distances, the soliton looks like a Dirac monopole [8], a classical solution of that theory [9]. The short-range interaction between the solitons is then seen as a result of screening, over the scale $\mu^{-1}$, of the long-range fields of these monopoles by polarization charges.

The two-dimensional version of the present theory may be applicable to a synthetic dielectric: the insulating phase of an array of Josephson junctions. (For an experimental study of the phase diagram of such an array, see Ref. [10].) In this case, the role of the polarization field is played by the dipole moments of the junctions, and the solitons are naturally interpreted as Cooper pairs. For this to be feasible, one should be able to quantize the solitons as bosons. Indeed, when we discuss the quantum version of the theory, we will

observe that the structure of the solitons allows them to be quantized as either fermions or bosons (or as anyons in two dimensions).

The paper is organized as follows. To make the presentation self-contained, we begin, in Sec. 2, with a review of the electron-as-soliton picture of Ref. [4]. In Sec. 4, we describe an analytical calculation that allows one to understand the mechanism responsible for screening the soliton interactions. We then proceed (in Secs. 5–7) to a discussion of quantum effects implied by the solitonic picture. The main outcome of this discussion is a curious version of the charge-vortex duality, one aspect of which is that quantization of vorticity, usually associated with superconductors, persists in dielectrics. Existence of quantized magnetic excitations results in a topological contribution to the magnetic susceptibility, associated with closed-path tunneling of polarization charges. In the ground state of a typical dielectric the effect is likely to be weak, but we suggest that the prospects for its observability may be improved by going to excited states, such as supplied, for instance, by Mie resonances in a small crystals. We summarize our results in Sec. 8. Some details of the calculations for two spatial dimensions appear in the Appendix.

## 2 Solitons in a nonlinear theory of dielectrics: a review

We begin with a review of the solitonic theory of electrons in a dielectric proposed in Ref. [4]. It employs, as its dynamical variable, the usual polarization vector $\boldsymbol{p}$ of macroscopic electrodynamics. This vector is defined on a lattice, and its lattice divergence determines the polarization charge density $\rho$ by the usual formula

$$\rho = -\nabla \cdot \boldsymbol{p} \, . \tag{1}$$

In application to a specific material, the lattice may be expected to correspond to that of the material, but for our present purpose (elucidation of the screening mechanism) it is sufficient to consider a simple cubic lattice with unit spacing and the primitive vectors oriented along $x$, $y$, and $z$. The polarization vector $\boldsymbol{p} = (p_x, p_y, p_z)$ is defined on the elementary faces (plaquettes) according to the following rule: a plaquette orthogonal to a primitive vector $\boldsymbol{n}$ hosts a single polarization component, $\boldsymbol{n} \cdot \boldsymbol{p}$ (see Fig. 1). We will assume that the components of $\boldsymbol{p}$, as well as the charge density $\rho$, are measured in units of $e/2\pi$, where $e$ is the electron charge.

To describe the dynamics, we seek a Lagrangian that depends on $\boldsymbol{p}$ and its time derivatives and is subject to an additional condition—invariance with respect to adding closed $2\pi$ strings. Adding a $2\pi$ string is defined as selecting a directed path on the the dual lattice (formed by the centers of the unit cells of the original) and changing the components of $\boldsymbol{p}$ at the plaquettes crossed by that path by $2\pi$. Physically, changing $\boldsymbol{p}$ by $2\pi$ at a plaquette corresponds to transporting a charge $e$ across that plaquette, as for instance when charges

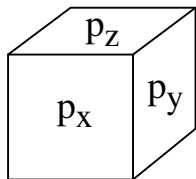

Figure 1: A unit cell of a simple cubic lattice, with each face hosting a single component of the polarization vector $\boldsymbol{p}$.

are separated through creation of a particle-hole pair. Pulling the charges further apart creates an open $2\pi$ string between them. The invariance with respect to adding closed strings means that the string has zero tension. As a consequence, the energy of the charges does not grow with their separation. This is precisely the rationale for requiring the invariance in question [4], as being able to separate charges to large distances without a persistent increase in energy should be the case for any dielectric.

## 2.1  Boundary conditions and the periodicity property

The invariance with respect to adding closed strings can be recast as a certain periodicity condition with the help of the Helmholtz decomposition of $\boldsymbol{p}$. The form of that decomposition is somewhat sensitive to the boundary conditions. In this section and the next, we adopt those that allow us to consider isolated solitons (i.e., do not restrict the total charge of the configuration to zero). They consist of the Neumann condition for the component $p_n$ of $\boldsymbol{p}$ orthogonal to the boundary ($b$) and the Dirichlet conditions for the components $\boldsymbol{p}_t$ tangent to it:

$$\partial_n p_n|_b = \boldsymbol{p}_t|_b = 0 \,. \tag{2}$$

For example, for those parts of the boundary that face the $x$ direction, we will use discretized versions of

$$\partial_x p_x|_b = 0 \,, \qquad p_y|_b = p_z|_b = 0 \,. \tag{3}$$

Physically, different boundary conditions correspond to different types of materials that the sample can interface with. The conditions (3) allow the electric current to flow in and out of the sample (the current density being proportional to $\partial_t \boldsymbol{p}$) and so are appropriate for an interface with a superconductor. Conditions appropriate for an interface with vacuum will be discussed in Sec. 6.

As we will see shortly, the boundary conditions (2) allow us to use the original Helmholtz decomposition,

$$\boldsymbol{p} = \nabla \times \boldsymbol{\psi} + \nabla \chi \,, \tag{4}$$

rather than Hodge's more general version. Here, $\boldsymbol{\psi}$ is a field that lives on the edges of the lattice (one component per edge, in accordance with the edge's direction), and $\chi$ is a scalar that lives on the sites of the dual lattice (centers of the unit cells). We use the continuum notation for the lattice derivatives. Thus, for example, the lattice curl of $\boldsymbol{\psi}$ is given by the circulation of $\boldsymbol{\psi}$ around a plaquette:

$$(\nabla \times \boldsymbol{\psi})_x(j, k, l) = \psi_z(j, k, l) - \psi_z(j, k-1, l) - \psi_y(j, k, l) + \psi_y(j, k, l-1), \qquad (5)$$

where triples of integers $(j, k, l)$ label both the plaquettes and the edges, and Eq. (5) indicates how these two types of labels are related. The discretization is chosen so that only $\chi$ contributes to the divergence in (1):

$$\nabla \cdot \boldsymbol{p} = \nabla^2 \chi, \qquad (6)$$

where $\nabla^2$ is the lattice Laplacian.

To argue that (4) indeed holds, we first define $\chi$, for a given $\boldsymbol{p}$, as the solution to (6) with the boundary condition $\chi|_b = 0$. We then have

$$\boldsymbol{p} = \boldsymbol{g} + \nabla \chi, \qquad (7)$$

where $\boldsymbol{g}$ obeys $\nabla \cdot \boldsymbol{g} = 0$ and the same boundary conditions as $\boldsymbol{p}$. We now observe that it can be written as $\boldsymbol{g} = \nabla \times \boldsymbol{\psi}$ for some $\boldsymbol{\psi}$ obeying the boundary conditions "dual" to (2):

$$\psi_n|_b = \partial_n \boldsymbol{\psi}_t|_b = 0. \qquad (8)$$

Explicitly, the components of $\psi$ in a specific gauge are

$$\psi_x(x, y, z) = \int_0^z g_y(x, y, z')dz' - \int_0^y g_z(x, y', 0)dy', \qquad (9)$$

$$\psi_y(x, y, z) = -\int_0^z g_x(x, y, z')dz', \qquad \psi_z(x, y, z) = 0, \qquad (10)$$

where the integration limits assume that three of the faces of the boundary coincide with the coordinate planes. The gauge arbitrariness inherent in the definition of $\boldsymbol{\psi}$ corresponds to transformations $\boldsymbol{\psi} \to \boldsymbol{\psi} + \nabla f$ with an arbitrary function $f$ satisfying the Neumann boundary conditions. We can use this for instance to go over to the gauge where $\nabla \cdot \boldsymbol{\psi} = 0$.

Next, we observe that any closed $2\pi$ string can be obtained as a composition of elementary $2\pi$ strings, each encircling one edge of the lattice. Adding such an elementary string amounts to changing $\boldsymbol{\psi}$ on the edge by $2\pi$. Since this does not affect the value of the energy, one can set up an equivalence relation with respect to such changes. This turns $\boldsymbol{\psi}$ into a compact gauge field—that of the lattice electrodynamics of Ref. [7].

## 2.2   Static energy functional

At the classical level, one can start by looking at static configurations. For these, the Lagrangian reduces to the negative of the static energy functional $E$:

$$L[\boldsymbol{p}, \partial_t \boldsymbol{p}] \to -\frac{1}{2\pi C} E[\boldsymbol{p}] \,, \tag{11}$$

where we have included a normalization coefficient in order to allow for arbitrary normalization of $E$.

The simplest representative enough form of $E[\boldsymbol{p}]$, satisfying the periodicity requirement, is

$$E[\boldsymbol{p}] = \sum_f V(\boldsymbol{p}) + \frac{1}{2} \sum_{cc'} (\nabla \cdot \boldsymbol{p})_c M_{cc'} (\nabla \cdot \boldsymbol{p})_{c'} \,, \tag{12}$$

where $V$ is a function that is $2\pi$ periodic in each of the components of $\boldsymbol{p}$, and $M_{cc'}$ are the elements of a symmetric matrix $\hat{M}$. The first term here is a sum over the elementary faces (plaquettes). As we will see, it represents finite polarizability of the medium. The second term is a sum over the unit cells. In view of the relation (1), it can be interpreted as an intrinsic capacitive effect, with $\hat{M}$ proportional to the matrix of inverse capacitances. We assume $\hat{M}$ to be positive definite.

Note that there are no terms involving transverse derivatives $\partial_m p_n$ with $m \neq n$: unlike $\nabla \cdot \boldsymbol{p}$, they are not invariant with respect to adding closed strings. In principle, we could have added periodic transverse terms such as $(\partial_m \sin p_n)^2$, as these do preserve the periodicity property. When $E[\boldsymbol{p}]$ is included in a Lagrangian supplying time-dependence, these terms give rise to dispersion of transverse optical phonons. Neglecting this dispersion, however, is not a bad approximation: it results in a sharp peak in the phonon density of state, which is indeed a feature of many common dielectrics (e.g., silicon [11]).

Consider a long open $2\pi$ string originating at some point of the dual lattice and extending to the boundary. By virtue of Eq. (1), the polarization charge density is nonzero only at the end of the string, with the total charge accumulated there being $2\pi$, or $e$ in physical units. Next, consider two opposite charges connected by a string. Since the string tension is zero, the energy does not depend on the length of the string. We can say that the interaction energy of the charges (prior to their coupling to electromagnetism) is zero. Due to the lattice derivative terms in $E$, however, this configuration is not a classical solution: we expect those terms to favor configurations where the charge density extends over some characteristic distance near the end points. In this way, the ends of open strings become solitons. The interaction energy is now nonzero and, while it stands to reason that it decays with the distance between the solitons, the precise form of this decay remains to be determined.

In the limit when the size of the soliton polarization cloud is large (in lattice units), the form of the functional (12) can be found by the derivative expansion. The first two terms

are

$$E[\boldsymbol{p}] = \sum_f V(\boldsymbol{p}) + \frac{1}{2}\sum_c (\nabla \cdot \boldsymbol{p})^2 \,. \tag{13}$$

We have used the arbitrariness of the coefficient in (11) to set the coefficient of the second term here to one half. Then, $C$ in (11) can be thought of as the self-capacitance of a unit cell, in appropriate units.

Throughout this paper, we assume cubic symmetry, meaning in particular that

$$V(\boldsymbol{p}) \approx \frac{1}{2}\mu^2 \boldsymbol{p}^2 \tag{14}$$

at small arguments. While the field $\boldsymbol{p}$ is not necessarily small at large distances from the soliton, the field $\tilde{\boldsymbol{p}}$, which is $\boldsymbol{p}$ with the string subtracted, is. Because $V(\boldsymbol{p})$ is periodic we can replace $\boldsymbol{p}$ in it with $\tilde{\boldsymbol{p}}$ and use the expansion (14) for $V(\tilde{\boldsymbol{p}})$.

Applicability of the derivative expansion requires $\mu \ll 1$ (in lattice units), the limit we refer to as an easily polarizable medium. In this case, the characteristic size of the polarization cloud is given by $\mu^{-1}$. In principle, there is no reason why this limit should hold for a particular dielectric. Without relying on the derivative expansion, we would have to consider the full linear operator corresponding to (12) at small $\tilde{\boldsymbol{p}}$,

$$\hat{\mathcal{L}} = -\nabla^2 + \mu^2 \hat{M}^{-1} \,. \tag{15}$$

As will be clear from the argument in Sec. 4, the behavior of the soliton fields at large distances is determined by that of the Green function of (15). In a system with translational invariance, this Green function can be written as

$$G(\mathbf{r}) = \sum_{\mathbf{k}} \frac{1}{\lambda_{\mathbf{k}}} e^{i\mathbf{k}\cdot\mathbf{r}} \,, \tag{16}$$

where the sum is over the Brillouin zone (BZ), and $\lambda_{\mathbf{k}}$ are the eigenvalues of (15). If we send the size of the lattice to infinity before taking the large $r = |\mathbf{r}|$ limit, we can replace the sum in (16) with an integral. Then, according to a well-known result of the Fourier series theory (see for example Sec. 12.D of Ref. [12]), if we model $1/\lambda_{\mathbf{k}}$ with a function analytic on the BZ, the Green function (16) decays exponentially at large $r$. In this manner, we can convince ourselves that our argument for screening is unaffected by going from (13) to the more general expression (12) (at least for a translationally invariant $\hat{M}$).

For clarity of the presentation, in the discussion of the screening mechanism in Sec. 4 we then proceed with the simpler expression (13). We have used it also for numerical work, results of which are presented in Sec. 3, with the parameter $\mu^{-1}$ there set typically just above the lattice scale ($\mu^2 = 0.1$). Discussion of observable effects in Sec. 7 does not rely on either of these choices. That discussion is somewhat sensitive, though, to the ratio

$$\alpha \equiv \mu^2/C \,, \tag{17}$$

in which $\mu$ appears in the Lagrangian (11). An estimate for this ratio will be given shortly.

We wish to stress that the soliton interaction discussed so far is an interaction of charges through the medium itself and, as such, applies prior to coupling of the system to electromagnetism. The relation (1), however, allows us to immediately construct an *additional* Hamiltonian describing the coupling to a static electric field. It reads

$$H_{el} = -\frac{e}{2\pi} \sum_c (\nabla \cdot \boldsymbol{p}) \Phi_{el} \,, \tag{18}$$

where $\Phi_{el}$ is the electrostatic potential. (We will discuss coupling to magnetic fields, which may be important for nonstatic states, in Sec. 6.)

Consider the sample in a static uniform electric field $\mathcal{E} = -\nabla\Phi_{el}$. For weak fields, combining (18) with the expansion (14), we find the equilibrium polarization to be

$$\boldsymbol{p} = \frac{e\mathcal{E}}{\alpha} \,, \tag{19}$$

where $\alpha$ is the parameter (17). In physical units, (19) corresponds to the polarization vector equal to $\mathbf{P} = e^2\mathcal{E}/(2\pi a\alpha)$, where $a$ is the lattice spacing. This allows one to relate the parameter $\alpha$ to the dielectric constant $\epsilon$ of the material [4]:

$$\alpha = \frac{2e^2}{a(\epsilon - 1)} \,. \tag{20}$$

For example, for silicon, using $a = 5.4\,\text{Å}$ and $\epsilon = 12$ [13], we find $\alpha = 0.48$ eV.

Restoring the full form of $V(\boldsymbol{p})$ will not be attempted here. It affects the energy of an individual soliton but not the interaction of solitons at large distances. We have used the simplest periodic form

$$V(\boldsymbol{p}) = \mu^2[1 - \cos(\boldsymbol{n} \cdot \boldsymbol{p})] \tag{21}$$

(the same as in Ref. [4]) for the numerical work but will not rely on this form in the discussion of observable effects in Sec. 7.

## 2.3 Restriction to two dimensions

It is of interest to consider the restriction of the present theory to two spatial dimensions. The counterpart of the energy functional (13), with $\boldsymbol{p}$ now defined on the edges of a square lattice with unit spacing, is

$$E = \sum_{\text{edges}} V(\boldsymbol{p}) + \frac{1}{2}\sum_c (\nabla \cdot \boldsymbol{p})^2 \,. \tag{22}$$

We expect this theory to apply to the insulating phase of an array of Josephson junctions, with the meaning of the parameters being especially clear in the case when the junctions are

realized as short superconducting wires (microbridges) connecting superconducting islands. For illustration, we will choose again $V(\boldsymbol{p})$ in the form (21), except that $\boldsymbol{n}$ is now the unit vector orthogonal to an edge. The parameter $C$ in (11) then represents the self-capacitance of a superconducting island, and the parameter $\alpha$, Eq. (17), the fugacity of quantum phase slips in the wires [14, 15]. (Solitons exist also in the one-dimensional version of the theory and, in that case, directly in the continuum, where they are none other than the solitons of the sine-Gordon model. Here, however, our focus is on the two and three-dimensional cases.)

The analog of (4) in two dimensions is

$$\boldsymbol{p} = \nabla \times \phi + \nabla \chi \,, \tag{23}$$

where $\phi$ is a scalar defined on the vertices, and $\nabla \times \phi = (\partial_y \phi, -\partial_x \phi)$. Adding an elementary closed string surrounding a vertex corresponds to the change $\phi \to \phi \pm 2\pi$. Equivalence relation with respect to such changes converts $\phi$ into an angular variable.

It follows from (23) that the field $\boldsymbol{\pi}$ orthogonal to $\boldsymbol{p}$, i.e., having components $\pi_m = -\epsilon_{mn}p^n$ (where $\epsilon_{xy} = -\epsilon_{yx} = 1$ and a sum over $n = x, y$ is implied), can be written as

$$\boldsymbol{\pi} = \nabla \phi - \nabla \times \chi \,. \tag{24}$$

Referring to Table 1, we see that this field can be viewed as the covariant derivative in the phase-only model of an equivalent superconductor, in the gauge where the vector field $\mathbf{A}$ is a pure curl, $\mathbf{A} = \nabla \times \chi$. Upon substitution of $p^m = \epsilon^{mn}\pi_n$ into (13), the latter becomes precisely the static energy functional of such a superconductor. In particular,

$$(\nabla \cdot \boldsymbol{p})^2 = (\nabla \times \mathbf{A})^2 \tag{25}$$

provides a Maxwell term for $\mathbf{A}$. The interaction between vortices in this model is well known to decay exponentially [16], an effect ordinarily attributed to screening of the interaction by the vector field $\mathbf{A}$. In Appendix A, we interpret this effect directly in terms of the scalar $\chi$, to reveal close parallels with screening of monopoles in the three-dimensional case.

# 3   Numerical results

Before we proceed to discussion of the screening mechanism, we present results of numerical solution of the Euler-Lagrange equations corresponding to the energy functional (13) with the potential (21). These equations read

$$-\nabla(\nabla \cdot \boldsymbol{p}) + \partial V/\partial \boldsymbol{p} = 0 \,, \tag{26}$$

and should be satisfied on each plaquette of the cubic lattice. The boundary conditions (3) allow configurations with a $2\pi$ string extending from the center of any unit cell to the boundary of the lattice.

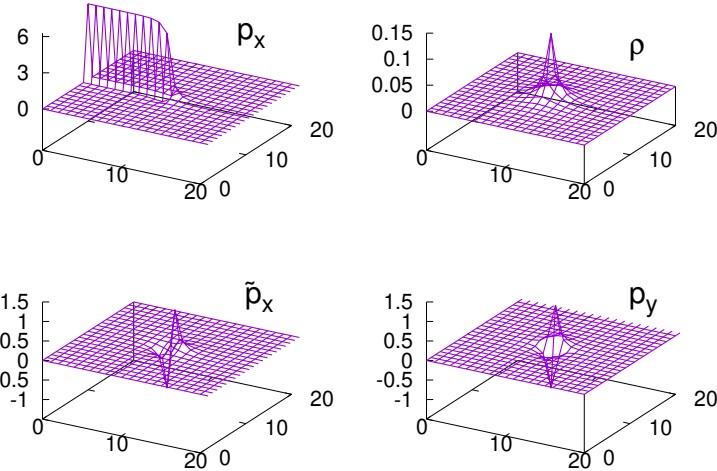

Figure 2: Profiles of the soliton field $\boldsymbol{p}$ and the density $\rho = -\nabla \cdot \boldsymbol{p}$ at the $(x, y)$ plane passing through the elementary ($q = 1$) soliton center in three dimensions, computed numerically on a $22^3$ grid for the potential (21) with $\mu^2 = 0.1$.

In Fig. 2, we show results obtained by application of the multidimensional Newton-Raphson method to Eq. (26). The soliton is located at the center of the grid, with the $2\pi$ string extending towards the negative $x$ direction. Note that, in addition to $p_x$, we plot the component $\tilde{p}_x$ of the field field $\tilde{\boldsymbol{p}}$, obtained by subtracting the string from $\boldsymbol{p}$: in the continuum notation,

$$p_x(\mathbf{r}) = \tilde{p}_x(\mathbf{r}) + 2\pi\Theta(x_0 - x)\delta(y - y_0)\delta(z - z_0), \tag{27}$$

$$p_y(\mathbf{r}) = \tilde{p}_y(\mathbf{r}), \tag{28}$$

$$p_z(\mathbf{r}) = \tilde{p}_z(\mathbf{r}). \tag{29}$$

where $\mathbf{r} = (x, y, z)$, $\Theta$ is the step function, and $(x_0, y_0, z_0)$ is the location of the soliton.

The story in two dimensions (2D) is very similar, with one exception. The latter concerns unstable solutions with multiples $q > 1$ of the elementary charge, by which we mean that they each have a string carrying $2\pi q$ of the electric flux. In three dimensions (3D), we have found such solutions for all $q \leq 5$. The ones with $q = 4$ and $5$ display a curious proliferation of maxima in the density profile, as illustrated for the case $q = 4$ in Fig. 3. There are total of six density maxima in this case (two of these lying off the plane of the plot). That is so even though the components of $\boldsymbol{p}$ themselves do not show any intricate structure, being similar in the overall shape to those of the elementary (stable) soliton. In 2D, on the other hand, we have found an unstable solution (with a single density maximum) for $q = 2$ but none for larger $q$.

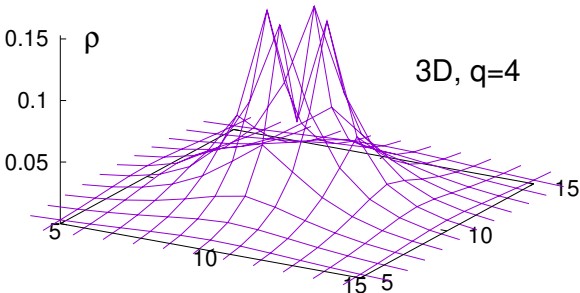

Figure 3: Same as in Fig. 2 but for the density profile of the unstable solution with charge $q = 4$. One can see four maxima of the density; there are two more off the plane, for the total of six.

# 4 The screening mechanism

We now describe an analytical method to compute the asymptotics of the fields and the soliton interaction at large distances. The strategy will be to assume (guided, say, by the numerical results of Sec. 3) that the field $\tilde{\boldsymbol{p}}$, obtained from $\boldsymbol{p}$ by subtracting the string, is small at large distances from the soliton center and argue that it is in fact small there *exponentially*.

Consider a single soliton centered at the origin with the string extending along the negative $x$ axis. We can absorb the string by a redefinition of the field $\boldsymbol{\psi}$ appearing in (4) into a new field $\tilde{\boldsymbol{\psi}}$, as follows:

$$\boldsymbol{\psi}(\mathbf{r}) = \boldsymbol{\kappa}(\mathbf{r}) + \tilde{\boldsymbol{\psi}}(\mathbf{r}), \tag{30}$$

where $\boldsymbol{\kappa}$ is the vector potential of a magnetic monopole. We will only need the monopole potential at large distances from the soliton center and so can use the expression for it in the continuum, found by Dirac [8]; in the Cartesian components, it reads

$$\kappa_x = 0, \quad \kappa_y = \frac{-z}{4r^2 \cos^2(\theta/2)}, \quad \kappa_z = \frac{y}{4r^2 \cos^2(\theta/2)}, \tag{31}$$

where $r = |\mathbf{r}|$, and $\theta$ is the polar angle measured from the positive $x$ direction. The tilded polarization vector, with the components given by (27)–(29), is then

$$\tilde{\boldsymbol{p}}(\mathbf{r}) = \frac{\mathbf{r}}{2r^3} + \nabla \times \tilde{\boldsymbol{\psi}} + \nabla \chi \equiv \boldsymbol{b} + \nabla \chi, \tag{32}$$

where we have defined a new field $\boldsymbol{b}$, an expression for which can be read off the above.

Replacing $\boldsymbol{p}$ with $\tilde{\boldsymbol{p}}$ does not change the periodic potential $V(\boldsymbol{p})$. Upon that replacement, the energy functional (13) becomes

$$E = \frac{1}{2}\sum_c (\nabla^2\chi)^2 + \sum_f V(\tilde{\boldsymbol{p}})\,, \tag{33}$$

where we have used Eq. (6). Consider the contribution to (33) from distances larger than some $R \gg 1$. Assuming that the magnitude of $\tilde{\boldsymbol{p}}$ is already small there, we can apply the expansion (14) to $V(\tilde{\boldsymbol{p}})$. Then, the contribution in question, in the continuum notation, is

$$\Delta E_R \approx \frac{1}{2}\int_{r>R} d^3x \left[(\nabla^2\chi)^2 + \mu^2\tilde{\boldsymbol{p}}^2\right]\,, \tag{34}$$

where according to (32)

$$\tilde{\boldsymbol{p}}^2 = (\nabla\chi)^2 + 2\boldsymbol{b}\cdot\nabla\chi + \boldsymbol{b}^2\,. \tag{35}$$

If we only want to use (34) as a basis for obtaining an equation for $\chi$ at large $r$, we can extend the integration there to all radii, but be prepared to encounter a singularity at $r = 0$ (where the expansion in small $\tilde{\boldsymbol{p}}$ does not apply). The singularity appears when we integrate by parts the term with the dot product: this produces $\nabla\cdot\boldsymbol{b}$ which, according to the expression for $\boldsymbol{b}$ from (32), equals $2\pi\delta(\mathbf{r})$. The equation for $\chi$ becomes

$$(\nabla^2)^2\chi - \mu^2\nabla^2\chi = 2\pi\mu^2\delta(\mathbf{r})\,. \tag{36}$$

The solution for $\nabla^2\chi$, which by (6) also gives the charge density, is immediate:

$$\nabla^2\chi = -\frac{\mu^2}{2r}e^{-\mu r}\,. \tag{37}$$

The total charge is

$$Q = -\int \nabla^2\chi\, d^3x = 2\pi\,, \tag{38}$$

independently of $\mu$. This corresponds to charge $e$ in physical units.

The form of the singularity on the right-hand side of (36) implies that, while $\nabla^2\chi$ is singular at $r = 0$, $\chi$ itself is regular there. So, when we solve Eq. (37) for $\chi$, we should include the $1/r$ solution to the homogeneous equation $\nabla^2\chi = 0$ with the coefficient chosen so as to remove the singularity. The result is

$$\chi(r) = \frac{1}{2r}(1 - e^{-\mu r})\,. \tag{39}$$

The same Eq. (34) can be used to see if the nonzero $\chi$ acts as a nontrivial source for $\tilde{\boldsymbol{\psi}}$. We will see shortly, using a slightly different method, that it does not. Assembling all the pieces of Eq. (32) together, we find that the solution for $\tilde{\boldsymbol{p}}$ at the linearized level is

$$\tilde{\boldsymbol{p}}(\mathbf{r}) = \frac{\mathbf{r}}{2r^3} + \nabla\chi = -\nabla\frac{e^{-\mu r}}{2r}\,. \tag{40}$$

We see that the gradient of the long-range $(1/r)$ part of $\chi$ precisely cancels the $\mathbf{r}/r^3$ term in (32). As a result, the fields $\tilde{\boldsymbol{p}}$ decay exponentially.

It is useful to consider also another representation of the vector (32), namely,

$$\tilde{\boldsymbol{p}} = \nabla \times \tilde{\boldsymbol{\psi}} + \nabla \tilde{\chi} \tag{41}$$

where

$$\tilde{\chi} = \chi - \frac{1}{2r} \,. \tag{42}$$

Using (41) in (34) and extending, as before, the integration to all radii, we obtain

$$\Delta E \approx \int d^3x \left\{ \frac{1}{2} [\nabla^2 \tilde{\chi} - 2\pi \delta(\mathbf{r})]^2 + \frac{\mu^2}{2} \left[ (\nabla \tilde{\chi})^2 + (\nabla \times \tilde{\boldsymbol{\psi}})^2 \right] \right\} \,. \tag{43}$$

The variational problem for $\tilde{\chi}$ thus amounts to finding a solution to

$$\nabla^2 [\nabla^2 \tilde{\chi} - 2\pi \delta(\mathbf{r})] - \mu^2 \nabla^2 \tilde{\chi} = 0 \,. \tag{44}$$

This is satisfied if we let $\nabla^2 \tilde{\chi} - 2\pi \delta(\mathbf{r}) = \mu^2 \tilde{\chi}$, meaning that the solution is none other than the suitably normalized Green function of the Helmholtz equation,

$$\tilde{\chi}(\mathbf{r}) = -\frac{e^{-\mu r}}{2r} \,. \tag{45}$$

Combining this with Eq. (42), we obtain again the solution (39) for $\chi$. This way of looking at the problem, however, underscores the generality of the screening mechanism. Indeed, we have already noted (in Sec. 2.2), that the argument for screening is unaffected if we replace (13) with the more general expression (12). We can see that by noting that this modifies the first term in (44) in such a way that, instead of (45), the solution is now proportional to the more general Green function (16). We also note that, according to (43), the field $\tilde{\boldsymbol{\psi}}$ is not sourced at the linear level: the variational equation for it is $\nabla \times (\nabla \times \tilde{\boldsymbol{\psi}}) = 0$, and the solution, in a suitable gauge, is $\tilde{\boldsymbol{\psi}} = 0$.

We find this screening effect remarkable, as it is quite distinct from the usual (Debye) screening of an external charge in a plasma. In that case, the induced charge density (the counterpart of our $\rho$) is $\rho_{ind} = -\frac{\mu^2}{4\pi} \Phi_{el}$, where $\Phi_{el}$ is the electrostatic potential, and $\mu^2$ is a coefficient depending on the parameters of the plasma. The Poisson equation $-\nabla^2 \Phi_{el} = 4\pi(\rho_{ext} + \rho_{ind})$ becomes

$$-\nabla^2 \Phi_{el} + \mu^2 \Phi_{el} = 4\pi \rho_{ext} \,. \tag{46}$$

For $\rho_{ext}$ corresponding to a unit point charge, the solution is

$$\Phi_{el}(\mathbf{r}) = \frac{e^{-\mu r}}{r} \,. \tag{47}$$

This solution is clearly different from (39). In particular, the total charge contained in it is zero, while that in (39) is not.

The physical pictures of the two effects are also completely different. This is most easily seen in the limit of a highly polarizable medium, when $\mu \ll 1$. Then, in the core region $1 \ll r \ll \mu^{-1}$, (47) is the potential of a point charge, surrounded at larger distances by a polarization cloud. The solution (39), on the other hand, is nearly constant at the core, so the core is nearly neutral. The fields $\boldsymbol{p}$ there are essentially those of the monopole of $\boldsymbol{\psi}$. The entire charge in this case is contained in a cloud of radius $r \sim \mu^{-1}$, apparently sourced by that neutral core.

On the other hand, one may note an analogy between the present mechanism and the screening of vortex lines in extreme type-II superconductors, where a nonmagnetic core sources magnetic flux. In this respect, the main result of this section is that a closely parallel mechanism exists for point-like (as opposed to line-like) defects in three dimensions.

The field $\tilde{\boldsymbol{p}}$ and charge density $\rho$ (the only objects on which the energy functional depends) decaying exponentially for a single soliton implies that so does also the soliton-soliton interaction. An explicit expression for it can be obtained in the limit $\mu \ll 1$. In this case, for a soliton and an antisoliton separated by a distance $L_s \gg 1$, the interaction energy accumulates mostly at distances $r \gg 1$ from the soliton centers, where we can use the quadratic approximation for $V(\tilde{\boldsymbol{p}})$ and work directly in the continuum. The result is

$$E_{int} = \int d^3 x \left[ (\nabla^2 \chi)_1 (\nabla^2 \chi)_2 + \mu^2 \tilde{\boldsymbol{p}}_1 \cdot \tilde{\boldsymbol{p}}_2 \right] , \tag{48}$$

where the subscripts 1 and 2 refer to the soliton and antisoliton, respectively. A direct computation gives

$$E_{int} = -\pi \mu^2 \frac{e^{-\mu L_s}}{L_s} . \tag{49}$$

For comparison, the energy of a single soliton (which always has to be computed on the lattice) is approximately linear in $\mu^2$ at small $\mu$, $E_{sol} \approx 4.68 \mu^2$, but deviates down from this line at larger $\mu$. For $\mu^2 = 0.1$, $E_{sol} = 0.42$.

# 5    Towards a quantum theory

So far, our considerations have been entirely at the classical level. To proceed to a quantum theory, we must include the static energy $E$, given by (13) or (22), as a part of a Lagrangian supplying the time dependence. The simplest choice, quadratic in time derivatives, of the Lagrangian in 3D is

$$L = \frac{1}{4\pi K} \sum_f (\partial_t \boldsymbol{p})^2 - \frac{1}{2\pi C} E , \tag{50}$$

where the sum is over the elementary faces, and $K$ and $C$ are constant coefficients. In the first term, $\boldsymbol{p}$ refers to the single component that the polarization vector has at each face. Note that $\partial_t \boldsymbol{p}$ is the polarization current density (in units of $e/2\pi$), so $K$ can be interpreted as the intrinsic inverse self-inductance per unit face, in appropriate units. [More generally, one can replace the first term with one including an inverse inductance matrix, similar to the inverse capacitance matrix $\hat{M}$ in (12).]

In 2D, the sum in (50) has to be replaced by a sum over the edges. We note that in this case the first term in (50) breaks the equivalence of our model with a phase-only superconductor, based on the transition from (23) to (24). As we will see, the model exhibits instead a curious version of self duality, with both electric and magnetic excitations present.

We can immediately identify two types of particle-like excitations in the theory (50). One is the soliton of the preceding sections, whose energy, upon including the normalization factor present in (50), is

$$E'_{sol} = E_{sol}/(2\pi C).\tag{51}$$

At the classical level, $E_{sol}$ is the dimensionless energy computed in the preceding sections; now, it may be expected to receive quantum corrections. The other type of excitation, which appears at the quantum level, is an optical phonon (a quantum of the polarization wave), with the characteristic energy $E_p = \hbar\sqrt{\alpha K}$, where $\alpha = \mu^2/C$ is the parameter defined in (17). We expect that an individual soliton can be considered classically when the ratio of these energies, $E_p/E'_{sol}$, is small. Since in our model the quantity $2E'_{sol}$ represents the electronic bandgap, this ratio is indeed small in a typical dielectric.

For silicon, using $E_p = 62$ meV, estimated from the peak in the phonon density of states [11], and $\alpha = 0.48$ eV already obtained from Eq. (20), and setting $\hbar = 1$, so that $K$ has the dimension of energy, yields $K = 8$ meV. Note that either estimate uses only the small-argument expansion (14) of the potential $V(\boldsymbol{p})$ and, as such, does not rely on the potential's precise form.

Even when an individual soliton can be well described classically, effects of the quantum statistics of solitons can become important at a finite density. In Fig. 2, we see that, while the charge density apparently respects the cubic symmetry of the model, the individual polarization components do not. This means that the rotational configuration space of the soliton is the full SO(3)in 3D or SO(2) in 2D. We note that the lattice pins the orientation of the soliton relative to the lattice directions, just as it pins its position to the center of a unit cell. This pinning, however, is only an energy barrier and does not affect the topology of the configuration space. Following the well-known argument [17], we then conclude that the solitons can be quantized either as bosons or fermions in 3D, or as anyons in 2D.

Conventional dielectrics will require quantization of the solitons as fermions. Quantization as bosons may be of interest in the context of a synthetic dielectric: an array of Josephson junctions in its insulating phase (see Ref. [10] for a study of the phase diagram

of such an array). In this case, the natural physical unit of charge, with respect to which we should require periodicity is $2e$ rather than $e$. Solitons of charge $2e$ can be interpreted as Cooper pairs.

Observation of a charge-$2e$ soliton in an array of Josephson junctions would be of interest in its own right and would also suggest that these systems may be useful for understanding the physics of conventional dielectrics, offering a degree of control over the parameters unavailable in the latter case. For conventional dielectrics, we focus in the following on novel magnetic excitations, which do not activate the charge subsystem at all and are thus insensitive to the statistics of the solitons.

# 6   Magnetic vortices in dielectrics

At the quantum level, the theory (50) has yet another type of excitation, one we have not discussed so far. This is the magnetic vortex. It is identified most readily by looking first at the 2D version. Let us use for that the Helmholtz decomposition (23) and proceed to the canonical quantization of the degree of freedom represented by $\phi$. The canonical momentum conjugate to $\phi$ is

$$\Pi = (2\pi K)^{-1}(-\nabla^2)\partial_t \phi \,. \tag{52}$$

Note that, just as $\phi$, $\Pi$ is defined on the vertices (sites) of the lattice. For reasons that will become clear shortly, we refer to it as vorticity. Because $\phi_v$ (on each lattice site) is a $2\pi$-periodic angular variable, each $\Pi_v$ is quantized in integer units.

The boundary condition for $\phi$ corresponding to (3) is $\partial_n \phi|_b = 0$, where $\partial_n$ is the normal derivative. Eq. (52) then implies a constraint (a global equivalent of the Gauss law) $\sum_v \Pi_v = 0$, where the sum is over all the vertices. As already discussed in Sec. 2.1, the boundary condition (2) allows for a flow of charge into the sample; it thus allows consideration of individual solitons. The above constraint, though, indicates that (2) does not allow consideration of individual quanta of vorticity (vortices). This may be understood by noting that it prohibits a nonzero density $\partial_t \boldsymbol{p}_t$ of the tangential electric current at the boundary, required to support an individual vortex. For this reason, it will be convenient for us to adopt in this section the boundary condition dual to the above, i.e.,

$$\phi|_b = 0 \,. \tag{53}$$

Eq. (53) prevents an inflow of charge but allows an inflow of vorticity. As such, this boundary condition is appropriate at an interface of the sample with vacuum. One can see that $\sum_v \Pi_v$ is now unconstrained.

Inverting (52) to find the current density $\nabla \times \partial_t \phi$ carried by $\phi$, we see that the eigenstate of $\Pi_v$ that has $\Pi_v = \delta_{v,v_0}$, i.e., $\Pi_v = 1$ at some $v = v_0$ and zero everywhere else, describes

a vortex pattern of current around $v_0$. We will refer to these vortices as magnetic, to distinguish them from the vortices at the cores of the electrically charged solitons considered in the preceding sections.

The kinetic term for $\phi$ in the Hamiltonian corresponding to (50) is

$$H_{kin} = \pi K \sum_v \Pi(-\nabla^2)^{-1}\Pi \,. \tag{54}$$

To understand its role, it is useful to consider first the limit when $K$ is the largest energy scale in the problem, namely, $K \gg \alpha$. As we have seen, this is not the case in a conventional dielectric such as silicon. Instead, as we will now argue, this limit bears hallmarks of a superfluid state. We may then expect it to be realizable in an array of Josephson junctions, where the phase diagram is known to include such a state (see, for example, Ref. [10]).

To argue the connection with superfluidity, we note that, in the limit $K \gg \alpha$, the term (54) is the leading term in the Hamiltonian of $\phi$, so the common eigenstates of all $\Pi_v$ are also approximate eigenstates of the Hamiltonian. Substituting

$$\Pi_v = \delta_{v,v_1} - \delta_{v,v_2} \tag{55}$$

in (54) we find that this term describes a logarithmic interaction between the magnetic vortices. This makes it clear that the vortices in question are none other than the usual vortices of a discrete 2D superfluid, with $K$ being proportional to the Josephson energy.

As one moves across the phase boundary to the insulating phase of the array, where $K \ll \alpha$, one can no longer rely solely on $H_{kin}$, as the second term in (50) becomes important. In this case, one may prefer to think of charges as solitons of the theory identical to the one described in Sec. 2, except possibly with a different value of the minimal charge (cf. the discussion in Sec. 5). On the other hand, the magnetic vortices can no longer be described semiclassically. We should use instead the fully quantum definition, with the addition of a vortex at site $v_0$ defined as the action of any operator $\mathcal{O}(v_0)$ such that

$$[\Pi_v, \mathcal{O}(v_0)] = \delta_{v,v_0}\mathcal{O}(v_0) \,, \tag{56}$$

where the square brackets denote a commutator.

Another way to arrive at the interpretation of the eigenstates of $\Pi$ as magnetic vortices is to consider interaction of our system with a vector potential $\mathbf{A}$. It is given, in parallel to (18), by the Lagrangian

$$L_{mag} = \frac{e}{2\pi c} \sum \mathbf{A} \cdot \partial_t \boldsymbol{p} \,, \tag{57}$$

where $c$ is the speed of light in vacuum, and the sum is over the plaquettes in 3D or the edges in 2D. Addition of (57) changes the canonical momentum of $\phi$ from (52) to

$$\Pi = (2\pi K)^{-1}(-\nabla^2)\partial_t\phi + (e/2\pi c)B \,, \tag{58}$$

where $B = (\nabla \times \mathbf{A})_\perp$ is the perpendicular magnetic field in units where the lattice spacing is set to one. This leads to the replacement $\Pi \to \Pi - (e/2\pi c)B$ in the kinetic term (54), which shows that a state with a nonzero expectation value of $\Pi_v$ carries a density of the magnetic moment.

In 3D, the counterpart of $\phi$ is the gauge field $\boldsymbol{\psi}$ of (4). Instead of the global symmetry $\phi \to \phi + \text{const}$, we now have the gauge symmetry $\boldsymbol{\psi} \to \boldsymbol{\psi} + \nabla f$. Accordingly, the constraint is now the usual Gauss law $\nabla \cdot \mathbf{\Pi} = 0$, where

$$\mathbf{\Pi} = \frac{1}{2\pi K}\nabla \times (\nabla \times \partial_t \boldsymbol{\psi}) + \frac{e}{2\pi c}\mathbf{B} \tag{59}$$

is the momentum conjugate to $\boldsymbol{\psi}$; just as $\boldsymbol{\psi}$ itself, it lives on the lattice edges. The second term here, with $\mathbf{B} = \nabla \times \mathbf{A}$ is due to the coupling (57). As usual in lattice gauge theories, the Gauss law represents conservation of flux at the vertices [18], so the magnetic vortices become vortex lines.

# 7   Prospects for observability

Quantization of vorticity in a dielectric is a somewhat unusual prediction of the present theory, so it is natural to ask what may be its observable consequences. We start by considering magnetic fluctuations in the ground state of a dielectric crystal. The simplest such fluctuation is spontaneous formation and disappearance of a localized magnetic moment. We will describe this process using a Euclidean classical solution (instanton) of the theory (50). The solution is purely magnetic, meaning that the charge field $\chi$ on it is zero. Upon continuation to the Euclidean time $\tau = it$, the equations of motion become

$$\partial_\tau^2 \boldsymbol{p} - \frac{K}{C}\frac{\partial V}{\partial \boldsymbol{p}} = 0\,, \qquad \nabla \cdot \boldsymbol{p} = 0\,. \tag{60}$$

Note that the first of these is a separate equation for the individual $\boldsymbol{p}$ on each plaquette. These $\boldsymbol{p}$, however, are connected by the second equation.

By virtue of the charge neutrality condition $\nabla \cdot \boldsymbol{p} = 0$, the capacitive terms, either those of the matrix type as in (12) or the simplified one as in (13), do not contribute to (60). Accordingly, the parameters $\mu^2$ and $C$ appear in (60) only through their ratio, $\alpha$, an estimate for which was given in Sec. 2.2.

We may use Eq. (60) without any changes also in the case when the crystal is placed in a time-independent magnetic field. The reason is that, for such a field, the additional Lagrangian (57) is "topological," in the sense that, being a perfect time derivative, it does not contribute to the classical equations of motion. At the quantum level, however, it can contribute a phase factor to a transition amplitude.

In a single instanton, one of the components of $\boldsymbol{\psi}$, say, $\psi_z$, changes from $0$ to $2\pi$ at a single edge during the Euclidean time interval $-\beta/2 < \tau < \beta/2$, where $\beta$ is a large parameter, while remaining zero at all the other edges. The corresponding $\boldsymbol{p} = \nabla \times \boldsymbol{\psi}$ is then nonzero only at the four plaquettes that meet at the edge in question. Thus, the instanton corresponds to a circular current, resulting from tunneling of polarization charge around that edge. One can verify that the charge neutrality condition $\nabla \cdot \boldsymbol{p} = 0$ is satisfied.

The Euclidean action of the instanton is the sum of four individual contributions, one from each of the four plaquettes in question. Two of these have $p_x = \pm\psi_z$, and the other two $p_y = \pm\psi_z$ (recall that there is only one component of $\boldsymbol{p}$ per plaquette). A potential $V(\boldsymbol{p})$ respecting the cubic symmetry has the same value, $V(\psi_z)$, on all four. Combining this with the contribution from the kinetic term in (50) and including the interaction (57), where $\mathbf{A}$ corresponds to a constant uniform magnetic field $B$ in the $z$ direction, we find the Euclidean action to be

$$S_E = \frac{1}{2\pi} \int d\tau \left\{ \frac{4}{2K}(\partial_\tau \psi_z)^2 + \frac{4}{C}V(\psi_z) - \frac{ieBa^2}{c}\partial_\tau\psi_z \right\} . \tag{61}$$

We have restored the lattice spacing $a$, so that $B$ is now in physical units.

One may note that the action (61) is formally identical to that of a short superconducting wire written in terms of the variable—the dipole moment—dual to the gradient of the phase of the order parameter between the wire's ends [14, 15]. The case $\alpha \gg K$, of interest to us here, corresponds to the regime where the wire is nearly insulating. In the present system, the equivalent "wire" is formed by the current circulating around a lattice edge.

The profile $\psi_z(\tau)$ of the instanton solution is determined by the Euler-Lagrange equation corresponding to (61),

$$\partial_\tau^2 \psi_z + (K/C)V'(\psi_z) = 0 \tag{62}$$

with the boundary conditions $\psi_z = 0, 2\pi$ at $\tau \to \pm\beta/2$ respectively. (As already noted, the topological term, proportional to $B$, does not contribute to the equation of motion.)

An antiinstanton is obtained from the instanton by reversing the direction of the current. This only affects the last term in (61). The integration over $\tau$ in that term can be trivially carried out, so the Euclidean action can be written as

$$S_E = S_0 \mp \frac{ie\Phi}{c} , \tag{63}$$

where the upper (lower) sign corresponds to an instanton (antiinstanton), $\Phi = Ba^2$, and $S_0$ is the real part of the action. In the ground-state partition function, given by the path integral of $e^{-S_E}$, each instanton thus contributes a phase factor $e^{ie\Phi/c}$, and each antiinstanton a factor $e^{-ie\Phi/c}$. This results in the ground-state energy, $E_{gs}(B)$, becoming $B$-dependent.

If instantons are relatively rare (as we may expect to be the case in the ground state of a typical dielectric), their contribution to the path integral can be computed in the dilute-gas

approximation. It leads to appearance in the partition sum of the factor

$$Z_{inst} = \sum_{n_+=0}^{\infty} \frac{(\bar{n}N_{tot}\beta)^{n_+}}{n_+!} e^{ie\Phi n_+/c} = \exp\left(\bar{n}N_{tot}\beta e^{ie\Phi/c}\right) , \qquad (64)$$

where $\bar{n}$ is the instanton density (the average number of instantons per edge per unit time), $N_{tot}$ is the total number of edges in the $z$ direction, and $\beta$ is the total Euclidean time duration. An individual term in the sum is the contribution from paths containing $n_+$ instantons. Antiinstantons contribute a similar factor but with the sign in front of $\Phi$ reversed. The product of the two factors,

$$Z_{inst}Z_{anti} = \exp\left[2\bar{n}N_{tot}\beta\cos(e\Phi/c)\right] , \qquad (65)$$

thus includes contributions of paths with arbitrary numbers of instantons and antiinstantons. Writing the partition function as $\exp(-\beta E_{gs})$, we obtain

$$E_{gs}(B) = -2\bar{n}N_{tot}\cos(eBa^2/c) + \text{const} \qquad (66)$$

for the ground-state energy. If the volume magnetic susceptibility $\chi_v$ is small (as will be the case here), it can be obtained by differentiating $-E_{gs}$ twice with respect to $B$ at $B = 0$ and dividing the result by $N_{tot}a^3$:

$$\chi_v = -\frac{2\bar{n}e^2a}{c^2} = -7.40 \times 10^{-6}(\bar{n}/\text{eV})(a/\text{Å}) \qquad (67)$$

(in CGS units).

For a large $S_0$, the instanton density $\bar{n}$ is suppressed by the semiclassical factor $e^{-S_0}$. In the ground state of a typical dielectric, this suppression is likely to be very strong. For example, in the case of the cosine potential (21), the solution to (62) is the soliton of the sine-Gordon model, and

$$S_0 = \frac{16}{\pi}(\alpha/K)^{1/2} . \qquad (68)$$

With the values of the parameters listed in Sec. 5 for Si, the resulting $e^{-S_0}$ is very small. One should keep in mind that, unlike the estimates in Sec. 5, the value of the instanton action depends on the precise shape of the potential $V(\boldsymbol{p})$. Since (21) is only a model chosen for illustration, the value (68) may in practice not be accurate enough. Still, a comparably strong suppression should be expected also for a more realistic potential, given that the susceptibility in question is ultimately a result of polarization charges tunneling between different atoms.

On the other hand, it is known that tunneling amplitudes often grow rapidly (exponentially) with the amount of excitation available in the initial state. To prevent that initial excitation from dispersing, it appears advantageous to consider, instead of a large uniform

sample, an array of small dielectrics (nanocrystals). Enhancement of the magnetic response of small dielectrics is well known in optics (for a review, see see Ref. [19]), where it is described as a resonance of Mie's classical scattering theory. It would be interesting to see if the topological dc susceptibility (67) can be sufficiently enhanced at such a resonance (of the crystal with external radiation) to allow for its experimental measurement.

One may notice that the use of small linear dimensions of a sample for control of the magnetic response in this proposal has some similarity with the use of a small superconductor—a Cooper-pair box (CPB) [20, 21]—for control of the electric charge. In this analogy, an enhanced topological susceptibility of an excited state of a dielectric would correspond to strong charge quantization, while the suppressed susceptibility in the ground state to weak charge quantization, i.e., a CPB in the transmon regime [22].

# 8    Conclusion

We thus arrive at a curiously complete instance of the charge-vortex duality, with a near perfect symmetry between the electric and magnetic charges. Each can be considered, under suitable conditions, either as elementary or as solitonic. This duality is characteristic of theories that obey a certain periodicity requirement—namely, the invariance of the static energy with respect to adding closed strings of integer electric flux. In the present paper, we have focused on the consequences of this periodicity for dielectrics, where two effects stand out.

One is the existence of solitonic electrically charged excitations, with a charge density possibly extending over several lattice volumes [4]. Here, we have developed an analytical understanding of the mechanism by which the total electric charge of unity is accumulated over a finite distance, starting from an uncharged core. (This is opposite to the usual Debye screening, which starts with a charged particle and results in an object of the overall charge zero.) We have also obtained analytical results for the interaction between the solitons, confirming that it is short-ranged (prior to their coupling to electromagnetism), which is essential for models of this type to be viable as the basis for classical simulations of the carrier dynamics. We have also argued that the structure of the solitons allows them to be quantized as either fermions or bosons (or anyons in 2D).

The second effect is related to quantization of vorticity. It amounts to a topological contribution to the magnetic susceptibility, associated with closed-path tunneling of polarization charges. In the ground state of a typical dielectric, the effect is likely to be very small. A possible route to enhancing it may be access to an excited state of a small crystal via a Mie resonance.

Finally, while experimental discovery of either effect in a conventional dielectric would be most interesting, one may also consider searching for them in a synthetic insulator, formed

by an array of Josephson junctions. In that case, one may be able adjust the parameters to one's advantage.

# A  Screening in two dimensions

As we already noted, in two dimensions, the change of variables from $\boldsymbol{p}$ to $\boldsymbol{\pi}$, Eq. (24), allows one to understand screening of the interaction between solitons in the present model as that between vortices in an equivalent superconductor, with the vector field of the latter given by $\mathbf{A} = \nabla \times \chi$. In this Appendix, we consider the screening mechanism in 2D using instead the same method as used in the main text for 3D, to underscore the parallels between the two cases.

In 3D, the key ingredient was the definition (30) of the field $\tilde{\boldsymbol{\psi}}$ that absorbs the vector potential of a monopole. In 2D, the Helmholtz decomposition is given by (23), and the analog of $\tilde{\boldsymbol{\psi}}$ is the field $\tilde{\phi}$ defined, in the continuum notation, by

$$\phi(x, y) = \arctan(y/x) + \tilde{\phi}(x, y) \,. \tag{69}$$

The first term is a vortex that carries a $2\pi$ string in the negative $x$ direction. The boundary conditions (2) correspond to $\partial_n \phi|_b = 0$ and the same for $\tilde{\phi}$.

The counterpart of (32) is

$$\tilde{\boldsymbol{p}}(\mathbf{r}) = \frac{\mathbf{r}}{r^2} + \nabla \times \tilde{\phi} + \nabla \chi \tag{70}$$

for the field $\tilde{\boldsymbol{p}}$, which is $\boldsymbol{p}$ with the string subtracted. Here $r = |\mathbf{r}|$ and $\nabla \times \tilde{\phi} = (\partial_y \tilde{\phi}, -\partial_x \tilde{\phi})$. The remainder of the argument of Sec. 4 is now transferred to the 2D case almost literally. The equation for $\chi$ at large $r$ is the same Eq. (36), except that the delta function is two-dimensional. The solution, determined up to a constant, $c$, is

$$\chi(r) = -K_0(\mu r) - \ln r + c \,, \tag{71}$$

where $K_0$ is the modified Bessel function of order zero. The charge density is

$$-\nabla^2 \chi = \mu^2 K_0(\mu r) \,, \tag{72}$$

corresponding to the total charge of $2\pi$.

To rewrite (70) similarly to (41), i.e., as

$$\tilde{\boldsymbol{p}} = \nabla \times \tilde{\phi} + \nabla \tilde{\chi} \,, \tag{73}$$

we define

$$\tilde{\chi} = \chi + \ln r - c \,. \tag{74}$$

The solution (71) corresponds to an exponentially decaying $\tilde{\chi} = -K_0(\mu r)$. An expression for the energy analogous to (43) is

$$\Delta E \approx \int d^2 x \left\{ \frac{1}{2}[\nabla^2\tilde{\chi} - 2\pi\delta(\mathbf{r})]^2 + \frac{\mu^2}{2}\left[(\nabla\tilde{\chi})^2 + (\nabla\times\tilde{\phi})^2\right] \right\}. \tag{75}$$

It can be used to show that $\tilde{\phi}$ is not sourced at the linear level. Indeed, the equation for $\tilde{\phi}$ following from it is $\nabla^2\tilde{\phi} = 0$, and the solution, after a suitable fixing of the inessential additive constant, is $\tilde{\phi} = 0$. Combining this with the result for $\tilde{\chi}$, we see that the fields $\tilde{\boldsymbol{p}}$ decay exponentially.

In Sec. 4, we contrasted the screening mechanism there to the Debye screening in a plasma. Here, a natural point of comparison is screening of an external charge in a linear 2D capacitive environment. An example of such an environment is an array of tunnel junctions, such as considered for example in Ref. [23]. The role comparable to $\chi$ is played in that case by the electrostatic potential $\Phi_{el}$, which obeys an equation analogous to (46). The solution again has zero total charge. One way to express the difference with the present case is to observe that, in the model of Ref. [23], $\Phi_{el}$ is the only dynamical variable. Here, on the other hand, we also have the angular variable $\phi$, capable of supporting point-like defects (vortices). In other words, the full set of variables in our model is a vector, $\boldsymbol{p}$, not a scalar.

In the limit $\mu \ll 1$, the interaction between a soliton and an antisoliton separated by a distance $L_s \gg 1$, can again be obtained from the continuum expression (49), except that the integral is now two-dimensional. The result is

$$E_{int} = -2\pi\mu^2 K_0(\mu L_s), \tag{76}$$

the same as the interaction (per unit length) between vortex lines in a type-II superconductor [16]. In Fig. 4, this is compared with numerical results obtained on a lattice. For computation of the energy of a single soliton, on the other hand, the continuum approximation can only achieve the logarithmic accuracy (at small $\mu$), as there is a large contribution from $r \sim 1$. For reference, the energy of a single soliton with $\mu^2 = 0.1$, computed numerically, is $E_{sol} = 0.814$.

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
