# Peer review of "Hidden order in dielectrics: string condensation, solitons, and the charge-vortex duality"

_SciPost Physics_

## Round 1 · Referee Report · Anonymous (Referee 1) · 2025-3-8

Strengths

1- The manuscript is written in an overall clear, high-quality English. 2- The article provides a mathematically rigorous description of electrons in a dielectric medium in terms of solitons, and of the arising screening mechanism. 3-The author dedicates a whole section to the discussion of experimental evidences of emergent quantized magnetic excitations, with some emphasis on a possible experimental platform for the measurement of these effects.

Weaknesses

1- On too many instances, the author either introduces notations or equations without definition, proper physical motivation or reference to the litterature. 3- The article lacks an Appendix.

Report

In this article, the author proposes a description of electrons in a dielectric medium in terms of interacting solitons. For this description to be valid, the solitons must interact only at short distances. This suggests that a shielding mechanism is at play to suppress Coulomb interactions at longer distances. A description of this mechanism is provided and, surprisingly, it exhibits properties that differ from the expected Debye picture. Finally, the author proceeds to quantize the solitonic image of the system and demonstrates the emergence of magnetic excitations in the form of quantized vortices. The article concludes with a discussion of the prospects for experimental measurement of these effects.

The results presented in this article and their insightful discussion by the author show, in my opinion, a high degree of relevance to the physics of dielectric media. The mathematical rigor of their derivation also contributes to the good readability of the article. My only criticism concerns the lack of physical motivation of certain equations, which are introduced without justification. Similarly, the author has sometimes made statements without supporting them. In this sense, the absence of an appendix providing further details and filling in these gaps is particularly detrimental.

Provided that the author corrects these flaws and introduces an appendix to their manuscript, I would support publication in SciPost Physics.

Requested changes

1- In the introduction, 4th line of the 2nd paragraph: "stings"-> "strings" 2- Could the author provide a justification for the form taken by Eq (2), especially the dependence in the divergence of the polarization field. 3- Could the author please explicitate why "the precise form of $V(p)$ is not tooimportant here", right before Eq (3). 4- In Eq (20) and (22)-(23), please define the function $K_n$. 5- Following Eq (23), the author claims that one can fix $\tilde\phi=0$, please provide a proof in appendix. 6-Please provide in appendix the full derivation of Eq. (32). 7- Justify the form taken by the action $S_E$ introduced in Eq. (43). 8- Provide a derivation of Eq (44).

Recommendation

Ask for minor revision

---

## Round 1 · Referee Report · Anonymous (Referee 2) · 2025-5-31

Strengths

1 – The manuscript works out an appealing theoretical description of electrons in a dielectric as solitons of the polarization field.

2 – It connects different branches of physics and is clearly written to a large extent.

Weaknesses

1 – The manuscript is not self-contained as it refers too often to Ref. [2], which is not published in a refereed journal but only available in the arXiv.

2 – It remains unclear why any of the discussed phenomena have not yet been seen experimentally. For instance, the manuscript lacks a realistic estimate of the parameter mu, which represents the spatial extent of the soliton.

Report

On the one hand, the manuscript presents an appealing analytical study how electrons in a dielectric are described as solitons of the polarization field. This method seems to be promising for computer simulations of the dynamics of excited states. At the quantum level, the theory has, in addition to the solitonic electric, elementary magnetic excitations, the quantization of which results in an additional topological contribution to the magnetic susceptibility. With this the manuscript connects different fields of physics.

On the other hand, the manuscript is partially elusive concerning possible experimental signatures of the conjectured description of electrons. Prospects for observing the proposed quantized magnetic vortices in a dielectric are discussed. But it remains unclear which order of magniture the screening parameter mu might have, which represents a key property of the underlying theory. And it is vaguely mentioned that solitons can be quantized either as bosons or fermions in 3D, or as anyons in 2D. Any discussion about how this conclusion affects the experimental detection is lacking.

Requested changes

1 – Elaborate further on the experimental observability.

2 – Make the manuscript self-contained by considering to merge the manuscript with unpublished Ref. [2].

Recommendation

Ask for major revision

---

## Round 2 · Referee Report · Anonymous (Referee 2) · 2025-11-9

Strengths

1 – The manuscript works out an appealing theoretical description of electrons in a dielectric as solitons of the polarization field.

2 - At the quantum level, the theory has, in addition to the solitonic electric, elementary magnetic excitations, which give rise to a topological contribution to the magnetic susceptibility.

3 - Prospects for observability are discussed.

Weaknesses

None

Report

During the resubmission process the manuscript was considerably improved. Not only the questions of the previous two reports were convincingly answered. But the whole manuscript was completely revised. It is now written in a coherent way, explaining carefully both its mathematial and physical content.

The manuscript easily meets two criteria of SciPost Physics. It provides a novel link between different research areas and it opens a new pathway in an existing research direction, with clear potential for follow-up work.

Due to the reasons mentioned above the manuscript should be published by SciPost Physics as it stands.

Requested changes

None

Recommendation

Publish (easily meets expectations and criteria for this Journal; among top 50%)

---

## Round 2 · Referee Report · Anonymous (Referee 1) · 2025-11-24

Report

In its current form, the novelty and relevance of this work are presented more clearly than in its previous version. Its relevance to research areas dealing with topological defects and solitons beyond dielectric media is also now more obvious, making it appealing to a broader readership.
In lights of the extensive efforts made by the author to improve the manuscript's readability and overall structure, I would recommend it for a publication in SciPost Physics.

Recommendation

Publish (meets expectations and criteria for this Journal)

---

## Round 2 · Author Response

Dear Editors,

I herewith resubmit my paper "Hidden order in dielectrics: ..." to SciPost Physics. In response to the referee comments, the paper has been reorganized and additional derivations have been included, as described below. Broadly speaking: (1) calculations related to screening in the two-dimensional case have been moved to the new Appendix A, which also includes additional technical details (as per a comment by Referee 1), while the main text now deals primarily with three dimensions; (2) the review section (Sec. 2) has been significantly expanded, to make the paper self-contained (as per a comment by Referee 2).

I have also expanded the Introduction, which now includes a table comparing the screening effect discussed in this paper to magnetic screening in superconductors.

The numbered changes requested by Referee 1 have been implemented as follows:

  1. The typo has been fixed (thank you!).

  2. A justification of the form of the static energy functional is now given in Sec. 2.2, starting from the more general expression (12) and including a general discussion of capacitive terms.

  3. Discussion of the form of V(p) (now around Eq. 21) has been rephrased.

  4. K_0 is now defined in Appendix A (where the 2D case has been moved).

  5. A derivation of \tilde{\phi} = 0 is also in Appendix A, after Eq. (75).

  6. A derivation of (32) (now Eq. 39) is now provided in the paragraph containing it.

  7. Justification of the form of S_E (now Eq. 61) is now given just above that equation.

  8. There is now a detailed discussion of the dilute instanton approximation (Eqs. 64, 65), leading to Eq. (66) (formerly Eq. 44).

The numbered changes requested by Referee 2 have been implemented as follows:

  1. The parameter \mu, while in principle important, does not affect the discussion of observability in Sec. 7 except as a factor in the parameter \alpha, Eq. (17). This is now stated just above Eq. (17). An estimate for \alpha is given shortly after that, following Eq. (20). Also, the effect discussed in Sec. 7, being magnetic rather than electric, does not depend on the statistics of the solitons. This is now stated explicitly in the last paragraph of Sec. 5.

  2. The paper has been made self-contained by expanding the review Sec. 2.

Sincerely, Sergei Khlebnikov

---

## Round 2 · List of Changes

Please see the resubmission letter for the list of changes.

---

## Editorial Decision

accepted_in_target_journal